# Antibiotic Prophylaxis or Granulocyte-Colony Stimulating Factor Support in Multiple Myeloma Patients Undergoing Autologous Stem Cell Transplantation

**DOI:** 10.3390/cancers13143439

**Published:** 2021-07-09

**Authors:** Eva-Maria Klein, Sandra Sauer, Sabrina Klein, Diana Tichy, Axel Benner, Uta Bertsch, Juliane Brandt, Christoph Kimmich, Hartmut Goldschmidt, Carsten Müller-Tidow, Karin Jordan, Nicola Giesen

**Affiliations:** 1Department of Medicine V, Hematology, Oncology and Rheumatology, University of Heidelberg, 69120 Heidelberg, Germany; sandra.sauer@med.uni-heidelberg.de (S.S.); uta.bertsch@med.uni-heidelberg.de (U.B.); juliane.brandt@med.uni-heidelberg.de (J.B.); kimmich.christoph@klinikum-oldenburg.de (C.K.); hartmut.goldschmidt@med.uni-heidelberg.de (H.G.); carsten.mueller-tidow@med.uni-heidelberg.de (C.M.-T.); karin.jordan@med.uni-heidelberg.de (K.J.); nicola.giesen@med.uni-heidelberg.de (N.G.); 2Department of Internal Medicine 5, Klinikum Nuremberg, Paracelsus Medical University, 90419 Nuremberg, Germany; 3Department of Infectious Diseases, Medical Microbiology, University Hospital Heidelberg, 69120 Heidelberg, Germany; sabrina.klein@med.uni-heidelberg.de; 4Division of Biostatistics, German Cancer Research Center (DKFZ), 69120 Heidelberg, Germany; d.tichy@dkfz-heidelberg.de (D.T.); benner@dkfz-heidelberg.de (A.B.); 5National Center for Tumor Diseases, 69120 Heidelberg, Germany

**Keywords:** multiple myeloma, autologous stem cell transplantation, anti-infective strategies, antibiotic prophylaxis, granulocyte-colony stimulating factor, infectious complications, multidrug resistant bacteria

## Abstract

**Simple Summary:**

Effective anti-infective strategies are of crucial importance in patients with multiple myeloma undergoing high-dose therapy followed by autologous stem cell transplantation (HDT/ASCT). We compare, for the first time, antibiotic prophylaxis versus granulocyte-colony stimulating factor (G-CSF) support as anti-infective strategies in this specific setting, including 353 individual cases of HDT/ASCT. We show similar efficacy in preventing infectious complications regarding antibiotic prophylaxis and G-CSF. Furthermore, we demonstrate that G-CSF support is associated with a shorter duration of inpatient stay and a lower rate of emerging multidrug resistant bacteria, especially vancomycin-resistant Enterococcus faecium. Therefore, G-CSF support should be the preferred anti-infective strategy in patients with multiple myeloma receiving HDT/ASCT.

**Abstract:**

We compare, in this manuscript, antibiotic prophylaxis versus granulocyte-colony stimulating factor (G-CSF) support as anti-infective strategies, in patients with multiple myeloma (MM), undergoing high-dose therapy followed by autologous stem cell transplantation (HDT/ASCT). At our institution, antibiotic prophylaxis after HDT/ASCT in MM was stopped in January 2017 and replaced by G-CSF support in March 2017. Consecutive MM patients who received HDT/ASCT between March 2016 and July 2018 were included in this single-center retrospective analysis. In total, 298 patients and 353 individual cases of HDT/ASCT were evaluated. In multivariate analyses, G-CSF support was associated with a significantly shortened duration of severe leukopenia < 1/nL (*p* < 0.001, hazard ratio (HR) = 16.22), and hospitalization (estimate = −0.19, *p* < 0.001) compared to antibiotic prophylaxis. Rates of febrile neutropenia, need of antimicrobial therapy, transfer to intensive care unit, and death, were similar between the two groups. Furthermore, antibiotic prophylaxis was associated with a significantly increased risk for the development of multidrug resistant bacteria especially vancomycin-resistant Enterococcus faecium compared to G-CSF support (odds ratio (OR) = 17.38, *p* = 0.01). Stop of antibiotic prophylaxis as an anti-infective strategy was associated with a reduction in overall resistance rates of bacterial isolates. These results indicate that G-CSF support should be the preferred option in MM patients undergoing HDT/ASCT.

## 1. Introduction

In patients with multiple myeloma (MM), infections are a leading cause of morbidity and mortality. Compared to the general population, the risk of developing an infection is increased seven-fold, attributable to the malignant disease itself and the administered therapy [1]. Disease-related risk factors and complications include an immunodeficiency mainly caused by a reduced production of polyclonal immunoglobulins and renal insufficiency or vertebral fractures [2]. In eligible patients, treatment with melphalan high-dose therapy, followed by autologous stem cell transplantation (HDT/ASCT), is considered a standard of care, as part of the first-line therapy [3,4]. Treatment-related toxicities after HDT/ASCT include neutropenia and mucositis with a high risk of febrile neutropenia, sepsis, and potentially death [2]. Due to improved supportive therapies, transplant-related mortality has been reduced to 1–2% in recent years [5].

The administration of antibiotic prophylaxis, mainly fluoroquinolones, following HDT/ASCT was shown to decrease the risk of febrile neutropenia and septicemia [6,7,8]. However, a large prospective randomized trial examining levofloxacin prophylaxis versus placebo in cancer patients with expected neutropenia longer than seven days after chemotherapy, demonstrated no significant effect of antibiotic prophylaxis on mortality [9]. Due to the high prevalence of fluoroquinolone-resistant Gram negative bacteria and the increased risk of acquiring multidrug resistant (MDR) bacteria, even during short-term antibiotic prophylaxis, the value of antibiotic prophylaxis in patients with MM receiving HDT/ASCT remains controversial [10,11].

The application of granulocyte-colony stimulating factor (G-CSF) after HDT/ASCT represents an alternative anti-infective strategy. G-CSF shortens the duration of neutropenia, but no consistent clinical benefit in this specific setting, in terms of development of febrile neutropenia, infections, and need of empirical antimicrobial therapy, could be demonstrated so far [12,13,14,15]. This is mirrored in very heterogeneous recommendations within international guidelines, including ASCO, ESMO, and EBMT [16,17,18,19].

To our knowledge, no study directly comparing antibiotic prophylaxis versus G-CSF support as anti-infective strategies, after HDT/ASCT in MM, has been published so far. Therefore, the aim of the present study was to compare antibiotic prophylaxis with ciprofloxacin or cotrimoxazole versus G-CSF support, with respect to blood count recovery, infectious complications, and emerging MDR bacteria in patients with MM, undergoing HDT/ASCT.

## 2. Materials and Methods

### 2.1. Anti-Infective Strategies

At our institution, a tertiary referral and transplant center, antibiotic prophylaxis with daily ciprofloxacin or cotrimoxazole after HDT/ASCT in MM was stopped in January 2017, and replaced by G-CSF support with filgrastim in March 2017. Prophylaxis against *pneumocystis jirovecii* pneumonia (PCP) with cotrimoxazole thrice weekly was implemented, beginning in March 2017. All patients received antiviral prophylaxis against herpes simplex and varicella zoster virus reactivation with acyclovir. Patients with prior hepatitis b infection (anti-HBc antibody positive) received antiviral prophylaxis with lamivudine, entecavir, or tenofovir. Detailed information about the anti-infective strategies during the period March 2016 to July 2018 are shown in Table 1. In the following, the anti-infective strategies are simplified to (I) antibiotic prophylaxis and (II) G-CSF support. A minor proportion of patients received (III) no prophylaxis during the change of the anti-infective strategy between January and March 2017.

### 2.2. Patient Cohort

Consecutive MM patients who received HDT with melphalan, 200 mg/m^2^, or dose-reduced in case of renal insufficiency followed by ASCT, as inpatients at our institution between March 2016 and July 2018, were included into this single-center retrospective analysis. Patients receiving HDT/ASCT as part of the first-line treatment or at relapse were included. In the case of tandem HDT/ASCT, both HDT/ASCT were evaluated separately. MM patients who were transplanted in an outpatient setting were excluded.

### 2.3. Data Collection

Data were collected by review of medical records regarding duration of leukocytes <1/nL, duration of platelets <20/nL, need of platelet transfusions, duration of inpatient stay, development of mucositis, febrile neutropenia and infections, detection of relevant pathogens, need of empirical or targeted antimicrobial therapy, new detection and type of MDR bacteria, transfer to intermediate or intensive care unit, death, hospital readmission after discharge, and response after ASCT. The duration of platelets <20/nL was defined by the time from the first day when platelets declined <20/nL to the first day when platelets rose ≥20/nL, regardless of administered platelet transfusions. The number of neutrophil granulocytes are not assessed at our institution in patients with leukocytes <1/nL. Therefore, the development of febrile neutropenia was defined by leukocytes <1/nL plus the occurrence of fever (ear temperature once ≥38.3 °C or ≥38.0 °C measured twice within twelve hours) [20]. Infections were further characterized as fever of unknown origin (FUO), respiratory infections, gastrointestinal infections, septicemia, sepsis, and other infections. Regarding antimicrobial therapies, the need of broad-spectrum antibiotics, carbapenems, antibiotics specifically effective against Gram-positive bacteria, reserve antibiotics, antimycotics effective against Aspergillus, and antivirals for the treatment of influenza were considered. The analysis was approved by the ethics committee of the University of Heidelberg (S-096/2017). All patients provided written informed consent.

### 2.4. Data Extraction of Bacterial Isolates and Resistance Rates in the Hematology Department and Transplant Unit between 2015 and 2019

Data regarding the absolute number of blood cultures sent for microbial analyses and the number of positive blood cultures, as well as the resistance rates to amoxicillin/clavulanate, cefuroxime axetil, ciprofloxacin, and cotrimoxazole were extracted from the database of the laboratory information system (Swisslab, Nexus AG, Berlin, Germany), retrospectively. All bacterial species tested for the respective antimicrobial substance were included in the analyses to calculate overall resistance rates. Equally, data were extracted regarding isolates and resistance rates from all tested materials.

### 2.5. Statistical Analyses

The collected data of the HDT/ASCT inpatient stay were analyzed regarding differences between the conducted anti-infective strategies using the chi-square test. Multivariable mixed effect models were applied to consider potentially repeated measurements due to tandem ASCT per patient. Multivariable mixed effect Cox regression analyses were conducted to assess the impact of the regimes (I)–(III) on the duration of leukocytes <1/nL and platelets <20/nL. The Cox proportional hazards model was applied to allow patients who were released before leukocytes were ≥1/nL or before platelets were ≥20/nL and, therefore, were censored at the end date of inpatient stay. Furthermore, there were no competing events during the observation period. The impact of the anti-infective strategies on the duration of inpatient stay was assessed by multivariable linear mixed effects regression analysis. Furthermore, a multivariable mixed effects logistic regression approach was applied to analyze the effect of the anti-infective strategies on the need of carbapenems, and a new detection of vancomycin-resistant *Enterococcus faecium* (VRE). The impact of the strategies (I)–(III) was adjusted for age at ASCT, ASCT at relapse versus ASCT in first-line treatment, response before ASCT, and transfused stem cell amount.

Cases of death within 30, 100, and 180 days after ASCT in the three groups of anti-infective strategy were assessed and respective mortality rates calculated.

A cost analysis was conducted examining the costs of an ASCT inpatient stay depending on the application of antibiotic prophylaxis or G-CSF. Because expensive drugs as reserve antibiotics were not needed during the inpatient stay in our study cohort and transfers to the intermediate or intensive care unit were similar in the groups, the cost analysis was simplified to the costs of antibiotic prophylaxis or G-CSF and the standard rate on general ward per day. These rates include especially costs for medical personnel, equipment and materials, food and laboratory examinations, and are around EUR 450 per day. For the costs of ciprofloxacin, cotrimoxazole, and G-CSF the purchase price our institution had to pay in June 2021 was used. For G-CSF, the dose for an adult weighing up to 70 kg was used. Costs of inpatient stay were determined as:(I)Antibiotic prophylaxis: median duration of inpatient stay in days x standard rate per day on general ward (EUR 450) + median duration of inpatient stay in days x costs ciprofloxacin/cotrimoxazole per day (EUR 0.12/EUR 0.14).(II)G-CSF support: median duration of inpatient stay in days x standard rate per day on general ward (EUR 450) + duration from ASCT to leukocytes ≥1/nL in days x costs G-CSF (EUR 5.75) + median duration of inpatient stay in days x cost cotrimoxazole (EUR 0.07)/2.

All results were assessed on the 5% significance level. R version 3.6.2 was used for performing the statistical analysis (www.r-project.org/ accessed on 15 April 2021). Figures regarding resistance rates of blood culture and all tested isolates were done with GraphPad Prism Version 5.0 (GraphPad, San Diego, CA, USA).

## 3. Results

### 3.1. Baseline Characteristics

In total, 298 patients with 353 HDT/ASCT were included in this analysis. The applied anti-infective strategies were (I) antibiotic prophylaxis in 151 HDT/ASCT and (II) G-CSF support in 150 HDT/ASCT. As antibiotic prophylaxis, ciprofloxacin was administered in 60 (39.7%), cotrimoxazole daily in 86 (57.0%), and another antibiotic prophylaxis in five (3.3%) cases. In 52 HDT/ASCT, (III) no prophylaxis was applied.

The median age at HDT/ASCT was 61 years. The standard melphalan dose of 200 mg/m^2^ was administered in >90% of patients in all groups. Further characteristics at HDT/ASCT in the entire cohort and in the groups (I)–(III) are shown in Table 2. Patient characteristics at diagnosis can be found in Appendix A.

### 3.2. Recovery of Blood Counts and Duration of Inpatient Stay

The median time from leukocytes <1/nL to ≥1/nL was 9 days in patients receiving (I) antibiotic prophylaxis compared to 6 days in patients receiving (II) G-CSF support. In multivariate analyses, G-CSF support was associated with a significantly shortened time to leukocytes ≥1/nL compared to antibiotic prophylaxis (*p* < 0.001, hazard ratio (HR) = 16.22, 95% confidence interval (95% CI) = 10.88–24.18; Table 3). In accordance, G-CSF shortened median duration of inpatient stay (*p* < 0.001, estimate = −0.19, 95% CI = −0.25–−0.12; Table 3). Furthermore, older age prolonged the duration of hospitalization. The time from platelets <20/nL to ≥20/nL did not differ between the anti-infective strategies (Appendix A). The median number of administered platelet transfusions was one in both groups.

### 3.3. Infections

The rate of febrile neutropenia was similar in the groups (I) antibiotic prophylaxis (130/151; 86.1%) and (II) G-CSF support (129/150; 86.0%). The most common infection in both groups was FUO ((I) 76/151; 50.3% versus (II) 80/150; 53.3%). Respiratory infections were slightly more frequent in patients receiving (I) antibiotic prophylaxis (15/151; 9.9%) compared to (II) G-CSF support (7/150; 4.7%). Rates of septicemia were similar ((I) 22/151; 14.6% and (II) 25/150; 16.7%) (Figure 1A). There was no statistically significant dependence between the groups of anti-infective strategy and the type of infection (*p* = 0.15). Mucositis was present in (I) 90.1% (136/151) and (II) 90.0% (135/150).

During the inpatient stay, a relevant pathogen was detected in blood, urine, or stool culture, bronchoalveolar lavage, or smear of the throat or another region in (I) 27.1% (41/151) and (II) 28.7% (43/150). The most common pathogens were *Escherichia coli*, *Clostridioides difficile*, *Pseudomonas aeruginosa*, influenza virus, and parainfluenza virus. In total, four patients developed PCP, two patients (2/236 HDT/ASCT; 0.85%) who had received prophylaxis with cotrimoxazole and two patients (2/117 HDT/ASCT; 1.71%) who had not received prophylaxis.

The need for empirical or targeted antimicrobial therapy was similar in the groups (I) antibiotic prophylaxis and (II) G-CSF support (Figure 1B). Multivariate analyses showed no difference regarding the need of carbapenems (odds ratio (OR) = 0.76, 95% CI = 0.32–1.80, *p* = 0.54; Table 4).

A transfer to the intermediate or intensive care unit was a rare event with (I) eight (5.3%) versus (II) nine (6.0%) patients. Two patients died during the inpatient stay, one in the group (I) antibiotic prophylaxis and one in the group (II) G-CSF support, leading to an overall fatality rate of 0.6%. One patient died due to pulmonary sepsis, the other patient due to sepsis and paralytic ileus.

### 3.4. Detection of Multidrug Resistant (MDR) Bacteria

In total, 31 cases of newly acquired colonization with MDR bacteria were detected during the inpatient stay of the 353 HDT/ASCT. Twenty-five cases of VRE were observed in rectal swabs. Three Gram-negative bacteria with combined resistance to third-generation cephalosporins and fluoroquinolones and one carbapenem-resistant Enterobacter cloacae were detected. Additionally, in two cases, more than one MDR bacteria (VRE plus multidrug-resistant *Pseudomonas aeruginosa* and VRE plus *Escherichia coli* with combined resistance to third-generation cephalosporins and fluoroquinolones) were observed. Three MDR bacteria led to a clinically relevant infection. Two patients developed septicemia due to *Escherichia coli* and one patient developed urinary tract infection due to VRE. In multivariate analyses, antibiotic prophylaxis was associated with a significantly increased risk for the acquisition of VRE compared to G-CSF (OR = 17.38, 95% CI = 2.24–134.68, *p* = 0.01; Table 5).

### 3.5. Hospital Readmission after Discharge

Between hospital discharge and the first regular follow-up outpatient visit at our institution, 28 patients had to be readmitted to the hospital. Hospital readmissions were twice as frequent in patients receiving G-CSF support compared to antibiotic prophylaxis (10.7% versus 5.3%). However, comparing the total duration of inpatient stay (duration of inpatient stay plus duration of a possible readmission), G-CSF support was still associated with a significantly shortened total duration of inpatient stay compared to antibiotic prophylaxis (estimate = −0.17, 95% CI = −0.24–−0.10, *p* < 0.001; Appendix A). Infection-related readmissions were similar between group (I) and (II) (4.7% versus 6.0%). Readmissions not related to infections in patients receiving G-CSF support were mainly due to nausea, loss of appetite, and general weakness.

### 3.6. Omitting Antibiotic Prophylaxis and G-CSF

During the change of the anti-infective strategy between January and March 2017, MM patients received (III) no prophylaxis after HDT/ASCT at our institution. The rate of febrile neutropenia was higher in group (III) no prophylaxis (50/52; 96.2%) compared to (I) antibiotic prophylaxis (130/151; 86.1%) and (II) G-CSF support (129/150; 86.0%). However, in multivariate analyses, the impact was not statistically significant (OR = 5.71, 95% CI = 0.82–39.67, *p* = 0.08). Patients receiving (III) no prophylaxis more frequently required antibiotic treatment, especially with carbapenems (34/52; 65.4% versus group (I) 69/151; 45.7% and (II) 65/150; 43.3%) (Figure 1B). Multivariate analyses revealed that the administration of carbapenems was more than five-fold as likely in patients who received no prophylaxis compared to antibiotic prophylaxis (OR = 5.64, 95% CI = 1.24–25.63, *p* = 0.03; Table 4). Furthermore, the risk for the acquisition of VRE was also increased in patients receiving no prophylaxis compared to G-CSF (OR = 10.75, 95% CI = 1.12–103.43, *p* = 0.04; Table 5).

### 3.7. Outcome

At the first regular follow-up outpatient visit (median 46 days after ASCT), response was ≥very good partial remission in (I) antibiotic prophylaxis 74.0%, (II) G-CSF support 74.5% and (III) no prophylaxis 76.9%. In total, six patients died within 180 days after ASCT. Within 30 days after ASCT, one case of death occurred in group (I) (mortality rate 0.66%). The 100-day mortality rate was (I) 0.66% (*n* = 1) and (II) 1.33% (*n* = 2). The 180-day mortality rate was (I) 0.66% (*n* = 1) and (II) 3.33% (*n* = 5). There were no cases of death in group (III). Due to low numbers of deaths, only descriptive analyses were conducted.

### 3.8. Results of the Cost Analysis

In our simplified cost analysis, the standard rate on general ward per day and the purchase price of ciprofloxacin, cotrimoxazole, and G-CSF were taken into account. Costs of empiric antibiotic therapy were not included because, due to expiry of patent protection, these drugs are not expensive anymore (e.g., purchase price of meropenem 500 mg EUR 1.57). Because transfers to the intermediate or intensive care units were similar in the groups, the associated costs were not considered as well as the costs of very rare hospital readmissions. In our simplified cost analysis, median costs of inpatient stay in patients receiving antibiotic prophylaxis with ciprofloxacin was EUR 8552 compared to EUR 8553 in patients receiving cotrimoxazole. The median cost per inpatient stay in the G-CSF support group was EUR 7708 €.

### 3.9. Resistance Rates of Bacterial Isolates in the Entire Hematology Department and Transplant Unit between 2015 and 2019

The resistance rates of blood culture isolates, as well as of all tested isolates to amoxicillin/clavulanate, cefuroxime-axetil, ciprofloxacin, and cotrimoxazole in the entire hematology department and transplant unit, are shown in Figure 2A,B. Overall, very high resistance rates were observed reaching >50% of tested isolates against all analyzed substances besides cefuroxime-axetil at all time points. In the period from 2015 to 2019, resistance rates against ciprofloxacin, which was stopped as antibacterial prophylaxis in the entire department outside of the allogeneic transplant setting at the beginning of 2017, continuously decreased and reached a plateau in 2018. In contrast, resistance rates against cotrimoxazole, which was equally stopped as antibacterial prophylaxis at the beginning of 2017, but then later that year widely reintroduced as PCP prophylaxis, reached a nadir in isolates from the year 2017, but increased to more than starting levels by 2019.

## 4. Discussion

To our knowledge, this is the first study comparing the anti-infective strategies—antibiotic prophylaxis versus G-CSF support after HDT/ASCT in a large cohort of MM patients. Antibiotic prophylaxis and G-CSF support showed similar rates of febrile neutropenia, pathogen detection, need of antimicrobial therapy, transfer to intermediate or intensive care unit, and death, reflecting equal efficacy in preventing infectious complications. In contrast, G-CSF support enables a lower rate of MDR bacteria and a shorter duration of inpatient stay, accompanied with lower costs.

At this time, there are very few studies comparing G-CSF support and antibiotic prophylaxis in cancer patients undergoing chemotherapy. Recently, a prospective randomized trial was presented at ASCO 2020, comparing ciprofloxacin prophylaxis with G-CSF support during adjuvant chemotherapy in patients with early stage breast cancer, showing lower rates of febrile neutropenia and a trend for reduced hospitalizations in patients receiving G-CSF [21]. A systematic review of studies investigating antibiotic prophylaxis vs. G (macrophage (M))-CSF support for the prevention of infections in cancer patients receiving chemotherapy found only two eligible randomized controlled trials, both showing no differences regarding febrile leukopenia, infections, and infection-related mortality, depending on the administration of antibiotic prophylaxis or G (M)-CSF [22,23,24]. The important question of new acquisition of MDR bacteria was not addressed within these trials.

There is no study available comparing antibiotic prophylaxis with G-CSF support in the context of HDT/ASCT in MM. Previously published studies show comparisons of antibiotic prophylaxis plus G-CSF vs. antibiotic prophylaxis or G-CSF and antibiotic prophylaxis vs. control [6,7,8,12,14,25]. In a prospective double-blind randomized trial examining moxifloxacin vs. placebo in patients with MM, lymphoma or solid tumor undergoing ASCT, antibiotic prophylaxis with moxifloxacin was associated with lower rates of bacteremia and a reduced duration of febrile episodes [6]. Furthermore, the recently published multicenter randomized TEAMM-trial demonstrated a reduction of febrile episodes and deaths in newly diagnosed MM patients receiving levofloxacin prophylaxis during first-line treatment [26]. Similarly, we show higher rates of febrile neutropenia in patients receiving no prophylaxis compared to antibiotic prophylaxis, accompanied by an increased need of carbapenems.

Our analysis suggests similar efficacy of antibiotic prophylaxis and G-CSF support in preventing infections. Previously published studies investigating antibiotic prophylaxis plus G-CSF support in MM patients receiving HDT/ASCT show lower rates of febrile neutropenia and septicemia, reflecting that this strategy might be even more effective [8,11]. However, other studies comparing antibiotic prophylaxis plus G-CSF support versus antibiotic prophylaxis show that adding G-CSF enables a shortening of neutropenia, but there is no consistent benefit along studies in preventing febrile neutropenia, septicemia, need of antibiotics, and death [12,15,27,28,29]. Whether the combination of antibiotic prophylaxis and G-CSF support leads to a real clinical benefit, also regarding emerging MDR bacteria, remains uncertain.

G-CSF support enables a shortening of the duration of severe neutropenia and hospitalization [12,13,15]. This was confirmed in our analysis. However, there was a trend for an increased risk of hospital readmission in the G-CSF support group with an emphasis on non-infection-related readmissions due to worsening of general condition, nausea, and loss of appetite. This may be explained by the fact that patients receiving G-CSF were discharged earlier from the hospital. Therefore, patients’ general conditions, food and fluid intake, and fitness levels should be assessed carefully before discharge. Nonetheless, when assessing the duration of inpatient stay plus the duration of a possible readmission, the total duration of inpatient stay was shorter in patients receiving G-CSF compared to antibiotic prophylaxis. This is of great clinical importance, because a shortening of hospitalization means preservation of resources and, especially, a gain in the quality of life of the patients.

Since the expiry of patent protection, prices for G-CSF have dropped significantly. Large hospitals being able to buy in bulk nowadays pay less than EUR 10 per daily dose of G-CSF for a normal weight adult. In contrast, hospital bed occupancy costs, including (especially) medical personnel, equipment and materials, food, etc., regularly run up to several hundred euros per day. Since highly expensive antimicrobial drugs, such as reserve antibiotics, were not required in our study cohort during the inpatient stay, and cost-intensive transferals to intermediate or intensive care units were very rare events occurring at similar frequency in the G-CSF and antibiotic prophylaxis groups, our cost analysis was simplified to assess the costs of inpatient hospital days plus administered G-CSF versus antibiotic prophylaxis. Our cost analysis demonstrates that the total costs of an inpatient stay are mainly determined by the duration of inpatient stay. Therefore, a shortening of inpatient stay due to application of G-CSF is accompanied with cost reduction.

We were further able to demonstrate that there is an increased risk of acquiring VRE in patients receiving no prophylaxis that is even similar to patients receiving antibiotic prophylaxis and might be explained by the high necessity of empirical or targeted antibiotic therapy. Infections caused by MDR bacteria constitute a worldwide threat. According to a recent analysis of the European Antimicrobial Resistance Surveillance Network data, there were 671,689 infections with antibiotic-resistant bacteria detected in countries of the EU and the European Economic Area in 2015, leading to 33,110 cases of death [30]. Compared to data from 2007, the cases of infections and associated deaths more than doubled [30]. Because the development of MDR bacteria is a result of a Darwinian selection process, particularly driven by misuse or overuse of antibiotics in human health care, animal husbandry, and agriculture, a careful consideration of antibiotic use is of utmost importance [31].

Our evaluation of resistance rates of bacterial isolates in the entire hematology department and transplant unit showed extremely high resistance rates, questioning the benefit of antibiotic prophylaxis. Furthermore, our analysis is able to confirm the association between antibiotic use and development of resistance. While omitting antibiotic prophylaxis with ciprofloxacin or cotrimoxazole led to a reduction of resistance rates, resistance against cotrimoxazole rose again after wide reintroduction as PCP prophylaxis.

In fact, in our analysis, G-CSF support reduced the risk of emerging MDR bacteria compared to antibiotic prophylaxis. Similarly, Maakaron et al. were able to demonstrate significant higher rates of MDR bacteria in MM patients receiving levofloxacin plus filgrastim after HDT/ASCT compared to lymphoma patients receiving only filgrastim after HDT/ASCT [11].

Antibiotic prophylaxis not only increases the risk of acquiring MDR bacteria, but also impacts the balance of our gut microbiota. Because there is an intensive interaction between our microbiota and immune system, a dysbiosis of the microbiota affects our immune system, too. For example, gut bacteria produce short-chain fatty acids that are able to down-regulate the NF-κB pathway, leading to an inhibition of tumor progression. Consequently, a dysbiosis of the gut microbiota due to antibiotic prophylaxis may impact response to cancer treatment and patient outcome [32]. In our cohort, response after ASCT was similar in the three groups of anti-infective strategy. Cases of death within 100 and 180 days after ASCT were rare overall, although more frequent in patients receiving G-CSF support compared to antibiotic prophylaxis. Due to the low number of fatal cases, no definitive conclusion as to any possible relationship with prophylaxis strategy can be drawn. However, as all deaths within the G-CSF group occurred later than 30 days after ASCT, a direct relationship seems unlikely.

Our study has some limitations, including its retrospective nature and consecutive minor heterogeneity regarding baseline characteristics between the groups of anti-infective strategies. Furthermore, HDT/ASCT without prophylaxis were conducted from January to March 2017 during influenza season, which might be a confounder. A major strength of our study, however, is the large cohort size of consecutively and uniformly treated patients.

## 5. Conclusions

To conclude, antibiotic prophylaxis and G-CSF support have similar efficacy in preventing infectious complications following HDT/ASCT in MM. Furthermore, G-CSF support enables a lower rate of emerging MDR bacteria and a shorter duration of inpatient stay, accompanied with cost reduction. Therefore, G-CSF support might be a preferable anti-infective strategy in MM patients receiving HDT/ASCT. Prospective randomized controlled trials are warranted to help identify optimal strategies to prevent infectious complications in these patients.

## Figures and Tables

**Figure 1 cancers-13-03439-f001:**
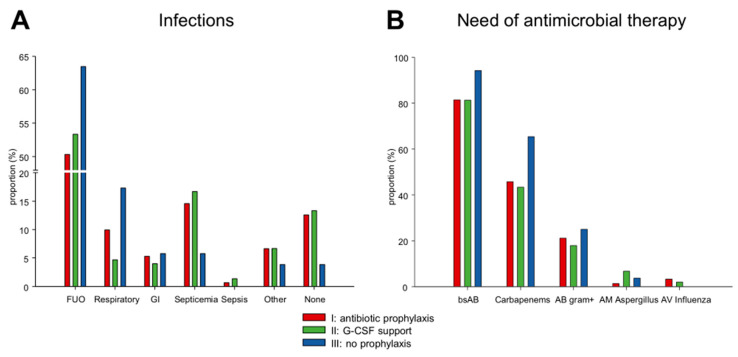
Frequency distribution of (**A**) infections and (**B**) administered antimicrobial therapies during the inpatient stay in the three groups of anti-infective strategy. Abbreviations: AB, antibiotics; AM, antimycotics; AV, antivirals; bsAB, broad-spectrum antibiotics; FUO, fever of unknown origin; GI, gastrointestinal.

**Figure 2 cancers-13-03439-f002:**
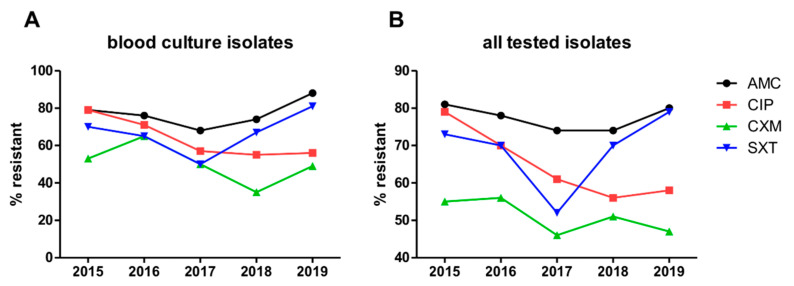
Resistance rates of (**A**) blood culture isolates and (**B**) all tested isolates in the entire hematology department and transplant unit between 2015 and 2019. Abbreviations: AMC, amoxicillin/clavulanate; CIP, ciprofloxacin; CXM, cefuroxime-axetil; SXT, cotrimoxazole.

**Table 1 cancers-13-03439-t001:** Summary of the anti-infective strategies (I) antibiotic prophylaxis, (II) G-CSF support, and (III) no prophylaxis.

Variable	(I) Antibiotic Prophylaxis	(II) G-CSF Support	(III) No Prophylaxis
Period A.D.	March 2016–January 2017	March 2017–July 2018	January 2017–March 2017
Ciprofloxacin 500 mg or cotrimoxazole 960 mg twice a day	yes	no	no
Filgrastim 5 μg/kg of BW daily until leukocytes > 2/nL	no	yes	no
PCP prophylaxis with cotrimoxazole 960 mg thrice weekly	not uniform *	yes	no
Acyclovir 400 mg twice a day	yes	yes	yes

* Patients receiving antibiotic prophylaxis with ciprofloxacin 500 mg twice a day had no PCP-prophylaxis. Abbreviations: BW, body weight; PCP, pneumocystis pneumonia.

**Table 2 cancers-13-03439-t002:** Characteristics of ASCT in the three groups of anti-infective strategy: (I) antibiotic prophylaxis; (II) G-CSF support; and (III) no prophylaxis.

Variable	(I)		(II)		(III)		All		*p*
	*n* (*n* = 151 ASCT)	%	*n* (*n* = 150 ASCT)	%	*n* (*n* = 52 ASCT)	%	*n* (*n* = 353 ASCT)	%	
Age at ASCT in years									0.06
Median (range)	59 (53–66)		62 (55–68)		62 (57–66)		61 (54–67)		
Response before ASCT									0.47
≥VGPR	85	56.3	83	55.3	34	65.4	202	57.2	
≤PR	60	39.7	63	42.0	17	32.7	140	39.7	
Not assessable	6	4.0	4	2.7	1	1.9	11	3.1	
Time point of ASCT									0.41
First-line	126	83.4	116	77.3	42	80.8	284	80.5	
Relapse	25	16.6	34	22.7	10	19.2	69	19.5	
Melphalan dose									0.37
200 mg/m^2^	138	91.4	143	95.3	49	94.2	330	93.5	
Other	13	8.6	7	4.7	3	5.8	23	6.5	
Stem cell amount									0.06
≥2.5 × 10 ^6^/kg of BW	123	81.5	109	72.7	45	86.5	277	78.5	
<2.5 × 10 ^6^/kg of BW	28	18.5	41	27.3	7	13.5	76	21.5	

Abbreviations: ASCT, autologous stem cell transplantation; BW, body weight; PR, partial remission; VGPR, very good partial remission.

**Table 3 cancers-13-03439-t003:** Results of the cox regression analysis on the impact of the anti-infective strategies (I)–(III) on the time from leukocytes <1/nL to ≥1/nL and results of the multivariate linear regression analysis on the duration of inpatient stay.

Variable	Time to Leukocytes ≥ 1/nL		Duration of Inpatient Stay	
	HR (95% CI)	*p*-Value	Estimate (95% CI)	*p*-Value
G-CSF support (vs. antibiotic prophylaxis)	16.22 (10.88–24.18)	<0.001	−0.19 (−0.25–−0.12)	<0.001
No prophylaxis (vs. antibiotic prophylaxis)	1.10 (0.72–1.69)	0.65	0.06 (−0.03–0.15)	0.16
Age (per ten years)	1.00 (0.82–1.21)	0.97	0.07 (0.04–0.11)	<0.001
ASCT at relapse (vs. first-line treatment)	0.86 (0.60–1.25)	0.44	0.06 (−0.02–0.13)	0.13
≥VGPR before ASCT (vs. ≤PR)	1.15 (0.85–1.55)	0.38	0.05 (−0.01–0.11)	0.12
Stem cell amount ≥ 2.5 * (vs. <2.5)	1.36 (0.94–1.96)	0.10	−0.07 (−0.14–0.00)	0.06

Abbreviations: ASCT, autologous stem cell transplantation; CI, confidence interval; HR, Hazard Ratio; S.E., standard error; VGPR, very good partial remission; * ≥2.5 × 10^6^ CD34^+^ cells per kg of body weight.

**Table 4 cancers-13-03439-t004:** Multivariable mixed effects model on the impact of the anti-infective strategy on the need of carbapenems.

Variable	Need of Carbapenems	
	OR (95% CI)	*p*-Value
G-CSF support (vs. antibiotic prophylaxis)	0.76 (0.32–1.80)	0.54
No prophylaxis (vs. antibiotic prophylaxis)	5.64 (1.24–25.63)	0.03
Age (per ten years)	1.20 (0.72–2.02)	0.48
ASCT at relapse (vs. first-line treatment)	2.08 (0.73–5.96)	0.17
≥VGPR before ASCT (vs. ≤PR)	1.40 (0.61–3.21)	0.42
Stem cell amount ≥ 2.5 * (vs. <2.5)	0.42 (0.14–1.23)	0.11

Abbreviations: ASCT, autologous stem cell transplantation; CI, confidence interval; G-CSF, granulocyte-colony stimulating factor; OR, Odds Ratio; PR, partial response; VGPR, very good partial response; * ≥2.5 × 10^6^ CD34^+^ cells per kg of body weight.

**Table 5 cancers-13-03439-t005:** Results of the multivariate logistic regression analysis on the impact of the anti-infective strategies (I)–(III) on the detection of vancomycin-resistant Enterococcus faecium (VRE).

Variable	Detection of VRE	
	OR (95% CI)	*p*-Value
Antibiotic prophylaxis (vs. G-CSF support)	17.38 (2.24–134.68)	0.01
No prophylaxis (vs. G-CSF support)	10.75 (1.12–103.43)	0.04
Age (per ten years)	1.53 (0.68–3.44)	0.30
ASCT at relapse (vs. first-line treatment)	2.79 (0.62–12.51)	0.18
≥VGPR before ASCT (vs. ≤PR)	2.12 (0.58–7.83)	0.26
Stem cell amount ≥ 2.5 * (vs. <2.5)	0.46 (0.11–1.99)	0.30

Abbreviations: ASCT, autologous stem cell transplantation; CI, confidence interval; OR; Odds Ratio; VGPR, very good partial remission; * ≥2.5 × 10^6^ CD34^+^ cells per kg of body weight.

## Data Availability

The data presented in this study are available upon request from the corresponding author. The data are not publicly available due to privacy issues.

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
