# Peer review of "Antibiotic Prophylaxis or Granulocyte-Colony Stimulating Factor Support in Multiple Myeloma Patients Undergoing Autologous Stem Cell Transplantation"

_cancers, 2021, doi:10.3390/cancers13143439_

Round 1
Reviewer 1 Report
Well written paper comparing the strategies of antibiotic prophylaxis, G-CSF support with no antibiotic prophylaxis and no antibiotic prophylaxis in autologous stem cell transplant recipients with MM.
Interesting findings of similar rates of febrile neutropenia, septicemia, empirical and targeted antibiotic use, mucositis and need for carbapenems between antibiotic ppx and G-CSF groups. The authors reported reduced duration of leucopenia and inpatient stay, with increased re-admits in G-CSF group due to non-infection related reasons such as poor nutrition and performance.
It would be nice to report the 30-day and 100-day mortality rates of the patients in the 3 groups to see the effects of the above findings [infections, MDR bacteria, poor nutrition, poor performance status etc] on the survival outcomes. Also, do you have data on the cost difference between the antibiotic ppx and G-CSF groups. Drugs in the flouroquinolones are much cheaper than G-CSF, so would be nice to know the cost-effectiveness aspect of these two different strategies. One could argue the hazard of MDR bacteria and the cost associated with it, hence a discussion on the financial aspects of these strategies is essential for it to be applied into routine clinical practice and is missing in this manuscript.
Good job exploring a novel concept in multiple myeloma patients!
Reviewer 2 Report
The topic and the study are controversial. Most centers around the world (to my knowledge) use both G-CSF and antibiotic prophylaxis for neutropenic infections. Reasons for not doing would be cost and selection of resistant microorganisms. This is not a randomized study, so other factors could play a role. The manuscript could be improved:
a) including the microbiome in their groups (if the authors collected samples)
b) make a detailed cost analysis of all drugs and days in the hospital
c) and most importantly- give data if the outcome is different in their 3 groups at 3 and 6 months (transition from PR- to CR respectively from VGPR to CR etc.)
Reviewer 3 Report
Dear. Dr. Klein
I have reviewed the manuscript entitled “Antibiotic prophylaxis or granulocyte-colony stimulating factor support in multiple myeloma patients undergoing autologous stem cell transplantation”. In this manuscript, you compared and examined anti-infective strategies antibiotic prophylaxis and granulocyte-colony stimulating factor (G-CSF) support in multiple myeloma patients undergoing high-dose therapy followed by autologous stem cell transplantation. The manuscript is well written, and the concept proposed by the authors is very interesting.
I have only minor comments.
- You showed similar efficacy in preventing infectious complications regarding antibiotic prophylaxis and G-CSF, and demonstrated various benefits of G-CSF. However, G-CSF has a very high cost compared with antibiotic prophylaxis. You should mention the cost-benefit about G-CSF.
- In table 1, the period among 3 groups is difficult to understand. I want you to add A.D.
- In table 2, are there any statistically differences of individual points among 3 groups and all cohort?
- In figure 1A, are there any statistically differences of FUO, Respiratory, Septicemia, sepsis, other, and None among 3 groups?
- What is the cause of death of two patients (line 253 - 255)?
Round 2
Reviewer 2 Report
Ok, my comments were addressed. However, I would strongly suggest to change this phrase in the abstract : We compare for the first time antibiotic prophylaxis.... to We compare in this manuscript antibiotic prophylaxis....
Having myself experience in myeloma for 30+ years and 20+ years in doing autologous transplants for MM, I am pretty sure this has studied before (maybe not in a randomized study....)
